# Lateral Control Calibration and Testing in a Co-Simulation Framework for Automated Vehicles

**Duc-Tien Bui \*, Hexuan Li** **, Francesco De Cristofaro and Arno Eichberger**

Institute of Automotive Engineering, Graz University of Technology, 8010 Graz, Austria;
hexuan.li@tugraz.at (H.L.); francesco.decristofaro@tugraz.at (F.D.C.); arno.eichberger@tugraz.at (A.E.)
\* Correspondence: d.t.bui@tugraz.at

**Abstract:** Lateral vehicle control is of high importance in automated vehicles as it directly influences the vehicle's performance and safety during operation. The linear quadratic regulator (LQR) controller stands out due to its high-performance characteristics and is used in the open source for self-driving functions. However, a notable limitation of the current approach is the manual calibration of LQR controllers based on the experience and intuition of the designers, leading to empirical uncertainties. To address this issue and enhance the lateral control performance, this paper concentrates on refining the LQR by employing three optimization algorithms: artificial bee colony optimization (ABC), genetic algorithm (GA), and particle swarm optimization (PSO). These algorithms aim to overcome the reliance on empirical methods and enable a data-driven approach to LQR calibration. By comparing the outcomes of these optimization algorithms to the manual LQR controller within an offline multibody simulation as a testing platform, this study highlights the superiority of the best-performing optimization approach. Following this, the optimal algorithm is implemented on a real-time system for the full vehicle level, revealing the model-in-the-loop and the hardware-in-the-loop gap up to 78.89% with lateral velocity when we use the relative error criterion (REC) method to validate and 2.35 m with lateral displacement when considering the maximum absolute value method.

**Keywords:** linear quadratic regulator; calibration optimization; virtual simulation; automated driving

## 1. Introduction

The advancement of highly automated vehicles holds a critical position in the field of automotive engineering. The key advantages of highly automated driving include improving road safety, particularly by minimizing driver errors and making efficient use of commuters' travel time [1,2]. Perception, trajectory planning, and control are three main parts in the structure of automated vehicles, where the control part allows the car to follow the trajectory, which has been determined in the trajectory planning parts. Because of the complex interconnections between the vehicle's lateral and longitudinal dynamics, designing a controller needs to be considered carefully [3] and it continues to remain a challenge. There are various control techniques that have been used for trajectory tracking in automated vehicles, such as: PID [4,5], linear quadratic regulator (LQR) [6–9], the sliding mode control (SMC) [10,11], robust control [12], model predictive control (MPC) [8,13–16] and reinforcement learning [17,18]. However, most of them are used to control longitudinal and lateral dynamics separately. Recently, controlling both longitudinal and lateral dynamics has been applied [4,19–23]. Nada et al. [21] designed a multi-input–multi-output (MIMO) linear MPC with some constraints in the vehicle dynamics, in which the reference path is tracked based on the steering angle and angular velocity. In [4], Zhou et al. adopted the MPC and nonlinear model predictive control (NMPC) method to cope with the nonlinear MIMO problem for the vehicle's lateral stability. Although the MPC controller can use nonlinear dynamics to solve the constraints, MPC

prevents having to conduct an experiment on a real-time system because of the increase in the model complexity and constraints as well as the huge amount of computation. The SMC technique is proposed to control by combining the lateral and longitudinal dynamic [22]. However, when applying the SMC technique, the chattering phenomenon often appears when acquiring robustness. To overcome this obstacle, in [23], the authors applied a new sliding mode that reduces chattering and makes state variables converge faster. However, rough road surfaces can lead to compromised path tracking accuracy and overall system stability.

The LQR approach is popularly used in automated vehicle controllers because of algorithmic simplicity, high-precision performance, and satisfaction with dynamic constraints. Zhang [24] compared the robust controller with LQR and MPC in terms of performance as well as computation. In the parking scenario, the LQR controller has a better performance. Regarding computation, the LQR requires less in comparison with MPC while it has the same performance indicators in some cases. However, LQR methods usually do not consider disturbances which can contribute to system errors. To cope with this problem, Kapania et al. [25] designed a controller by combining feedback and feedforward steering. The aim is to keep the stability of the vehicle under hard maneuvering conditions as well as minimize path deviation. However, the steady-state path deviation drastically increases at high velocity. In [26], the authors also proposed an LQR controller using feedforward and predictive steering for lateral dynamics, which helps vehicle driving under complex conditions. Although LQR controllers were widely applied to automated vehicles, most LQRs choose weight factors based on empirical consideration. To dispose of empiricism, some studies added algorithms to optimize the LQR controller in order to improve performance. In [27–29], the authors used GA, fuzzy control, and PSO to choose the optimized weight factors for the LQR controller. The results showed the effectiveness of these algorithms regarding the tracking accuracy and the stability of the vehicle. Although promising results are shown in these studies, there is still potential for further improvement in the accuracy of the controller.

On the other hand, because the requirement for automated vehicles level is higher, the scenarios for testing become complex. Consequently, novel challenges to the reliability of automated vehicles emerge in testing and validation. Additionally, the growing need for testing and validation in terms of advanced driver assistance systems (ADASs) as well as automated driving technologies arises because real-world road scenarios have covered more and more [30,31]. Consequently, X-in-the-Loop (XiL) has emerged and become a predominant approach for the scenario-driven simulation testing of automated vehicles. The "X" expressed the various development focuses: model, software, processor, hardware, vehicle, and driver. Hye Young An [32] proposed a path planning algorithm and pure pursuit in real-time to control the vehicle by considering the detected lanes and some constraints. The authors only considered a single lane and the error is approximately 1.147 m when the car exits the roundabout. Taekgyu Lee et al. [33] employed a DNN-based controller to control the car. This method reduces the computational load in comparison with the previous NMPC method. In [34,35], the authors also used MPC and NMPC to control automated vehicles in the real-time system, and the results show the efficiency, robustness, as well as the feasibility of these methods. In [36], the authors used a low-level MPC to control the small-scale race cars. Based on the simulation and experimental findings, it appears that opting for a more cautious approximation with the discriminating kernel leads to a safer driving style. Most of the previous works did not compare the gap between office simulation and real-time system. This happened because of the differences in purpose, complexity, resources, and constraints.

The key contributions of this study are:

- Three optimization algorithms, namely artificial bee colony, particle swarm optimization, and genetic algorithm are implemented to find the best coefficient of the LQR controller. The primary objective is to eliminate the external disturbances arising from the desired trajectory. The algorithm optimizations are simulated on CarMaker

11.1 and Matlab/Simulink software 2021b. The effectiveness of three algorithmic enhancements is compared to the LQR controller performance without using them. Afterward, the results obtained from three algorithm optimizations are compared together to choose the best algorithm for the model-in-the-loop (MiL) simulation.

- The best algorithm for the MiL simulation is used to simulate a real-time system to assess the performance. The chosen algorithm is simulated on MiL and hardware-in-the-loop (HiL). The outcomes reveal the gap between the MiL simulation and HiL for the vehicle model under consideration.

The remainder of this study is structured as follows: Section 2 contains the vehicle dynamic of the model, the framework of Apollo, as well as the optimization. In Section 3, the model and controllers are simulated in an office environment and the real-time system using the multibody software CarMarker. Section 4 discusses the results and the Section 5 summarizes the research.

## 2. Vehicle Dynamics and Control

Apollo is a widely recognized open source SAE Level 4 AD platform that was introduced by the Baidu company [37]. It includes a comprehensive suite of hardware and software solutions for various aspects of AD, including perception, planning, and control.

### 2.1. Vehicle Model

Figure 1 depicts the vehicle dynamic model as:

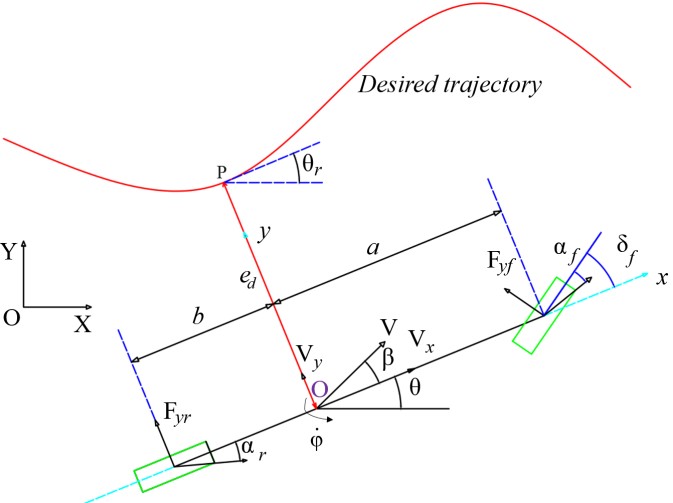

**Figure 1.** Vehicle dynamic model.

In this model, we assume that the model is simplified as the lateral dynamic model with two degrees of freedom, and vertical movement is ignored since the vehicle drives in a single plane. $\alpha_f$, $\alpha_r$ are steering angles in the front and rear wheels, respectively. The front and rear steering angles are small and noted by $\delta_f$ and $\delta_r$, so the lateral force and side slip angle has a linear relationship, which is acceptable for tire slip angles of about 3 degrees. The lateral load transfer effects which occur due to lateral acceleration are neglected and the suspension system's effect is not considered.

$O$ is the mass center point, $a$ and $b$ are the distances from $O$ to the front and rear axles, $\varphi$ is the yaw angle, and $\dot{\varphi}$ is yaw rate. $C_f$ and $C_r$ are the lateral stiffness of the tires in the front and rear wheels, $F_{yf}$ is the lateral force of the front wheel, and $F_{yr}$ is the lateral force of the rear wheel. $v_x$, $v_y$ are vehicle's longitudinal and lateral velocity, respectively.

The dynamic equations of this model are noted as [38]:

$$ma_y = F_{yf} + F_{yr} \tag{1}$$

$$I_z\ddot{\varphi} = aF_{yf} - bF_{yr} \tag{2}$$



$$\dot{v}_x = v_y \dot{\varphi} + a_x \tag{3}$$

where $m$ is the mass of the vehicle, $I_z$ is the rotation inertia of the vehicle around the vertical axis. All of these symbols are listed in the Nomenclature part.

Because the front wheel steering angle is small, the lateral force and side slip angle have a linear relationship, and the dynamic Equations (1) and (2) are obtained as follows:

$$
\begin{cases}
\dot{v}_y = \dfrac{C_f + C_r}{mv_x} v_y + \left( \dfrac{aC_f - bC_r}{mv_x} - v_x \right)\dot{\varphi} - \dfrac{C_f}{m}\delta \\
\ddot{\varphi} = \dfrac{aC_f - bC_r}{I_z v_x} v_y + \dfrac{a^2 C_f + b^2 C_r}{I_z v_x} \dot{\varphi} + \dfrac{aC_f}{I_z}\delta \\
\dot{v}_x = \dot{y}\dot{\varphi} + a_x
\end{cases} \tag{4}
$$

When the vehicle follows a reference path, a lateral error and heading angle error will occur. As illustrated in Figure 1, the lateral error, denoted by $e_d$, represents the shortest distance between the point $O$ and the projected point $P$ on the reference path. Meanwhile, the heading angle error, referred to as $e_\theta$, is the difference between the actual heading angle of the vehicle $\theta$ and the reference heading angle $\theta_r$. For the sake of simplicity, the side slip angle is assumed to be $\beta = 0$ at the point $O$, and the heading angle error is calculated as $e_\theta = \theta - \theta_r$. In practical control, the controller's task is to promptly eliminate these two errors in real-time to ensure that the vehicle stays on track with the planned path. With these errors in mind, it becomes possible to compute the first-order derivatives of the lateral error $\dot{e}_d$ and the heading angle error $\dot{e}_\varphi$:

$$\dot{e}_d = v_x e_\varphi + v_y \tag{5}$$

$$\dot{e}_\varphi = \dot{\varphi} - \dot{\varphi}_r \tag{6}$$

Equations (5) and (6) are substituted in Equation (4), which can obtain:

$$\ddot{e}_d = \frac{C_f + C_r}{mv_x}\dot{e}_d - \frac{C_f + C_r}{m}e_\varphi + \frac{aC_f - bC_r}{mv_x}\dot{e}_\varphi + \left(\frac{aC_f - bC_r}{mv_x} - v_x\right)\dot{\theta}_r - \frac{C_f}{m}\delta_f \tag{7}$$

$$\ddot{e}_\varphi = \frac{aC_f - bC_r}{I_z v_x}\dot{e}_d - \frac{aC_f - bC_r}{I_z}e_\varphi + \frac{a^2 C_f + b^2 C_r}{I_z v_x}\dot{e}_\varphi + \frac{a^2 C_f + b^2 C_r}{I_z v_x}\dot{\theta}_r - \frac{aC_f}{I_z}\delta_f \tag{8}$$

Equations (7) and (8) can be rewritten as:

$$
\begin{Bmatrix} \dot{e}_d \\ \ddot{e}_d \\ \dot{e}_\varphi \\ \ddot{e}_\varphi \end{Bmatrix} =
\begin{Bmatrix}
0 & 1 & 0 & 0 \\
0 & \dfrac{C_f + C_r}{mv_x} & -\dfrac{C_f + C_r}{m} & \dfrac{aC_f - bC_r}{mv_x} \\
0 & 0 & 0 & 1 \\
0 & \dfrac{aC_f - bC_r}{I_z v_x} & -\dfrac{aC_f - bC_r}{I_z} & \dfrac{a^2 C_f + b^2 C_r}{I_z v_x}
\end{Bmatrix}
\begin{Bmatrix} e_d \\ \dot{e}_d \\ e_\varphi \\ \dot{e}_\varphi \end{Bmatrix} +
\begin{Bmatrix} 0 \\ -\dfrac{C_f}{m} \\ 0 \\ \dfrac{aC_f}{I_z} \end{Bmatrix}\delta_f
$$

$$
+ \begin{Bmatrix} 0 \\ \dfrac{aC_f - bC_r}{mv_x} - v_x \\ 0 \\ \dfrac{a^2 C_f + b^2 C_r}{I_z v_x} \end{Bmatrix}\dot{\theta}_r \tag{9}
$$

Equations (9) can be rewritten in the state-space representations as:

$$\dot{X} = AX + BU + C\dot{\theta} \tag{10}$$

with: $X = [\dot{e}_d, \ddot{e}_d, \dot{e}_\varphi, \ddot{e}_\varphi]$ is the state vector; $U = [\delta_f]$ is the control input.

### 2.2. Lateral Controller

In this study, the path planning process involves a series of reference points. This discretization of data is necessary for practical implementation. To control the vehicle along this discrete trajectory, a discrete linear-quadratic regulator (dLQR) controller is used. Equation (10) is discretized to design the dLQR controller, which governs the vehicle's dynamics. During this process, we neglect the effect of $C_{\theta_r}$ and apply the midpoint Euler and the forward Euler approach to clarify the model while preserving essential characteristics. As a result, we obtain the equation that describes the discrete tracking errors as:

$$\dot{X}_{k+1} = \bar{A}X + \bar{B}U \tag{11}$$

with:

$$\bar{A}_k = (I - \frac{A\Delta t}{2})^{-1}(I + \frac{A\Delta t}{2})^{-1}; \bar{B} = B\Delta t \tag{12}$$

Figure 2 shows the entire structure of the LQR controller for an automated vehicle, which includes four main parts: the perception, path planning, controller, and the vehicle model. In path planning, the EM planner [39] is used to generate the reference path and path tracking errors. As the path planning module is not the focus of our work, the exact algorithm can be referenced in the GitHub project [40].

The LQR controller is the main center of the second part. The aim of this study is to find the optimized matrix K for the LQR controller using three algorithm optimizations (ABC, GA, and PSO) as well as calculate $\delta_{ff}$ in the feedforward control step to find the final signal control (steering angle $\delta_f$). Then, the steering angle $\delta_f$ is sent to the vehicle model to control a car. We use the BMW5 car model and IPG Carmaker 11 software as a simulation environment. After comparing the performances of the vehicle when using the three optimization algorithms, we find the best algorithm for the car. Finally, we use this algorithm to simulate the real-time system and compare the gap results between the MiL and HiL.

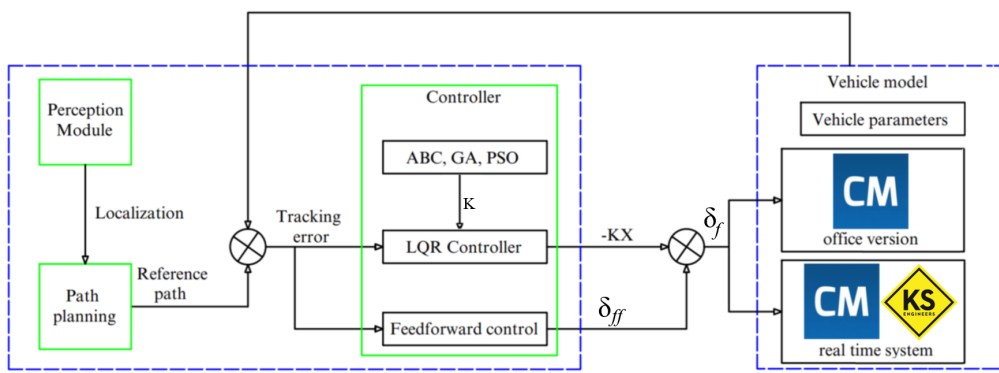

**Figure 2.** Lateral control structure.

The cost function of LQR controller is defined as:

$$\sum_{0}^{\infty}(X_k^T Q X_k + U_k^T R U_k)dt \tag{13}$$

$$Q = diag(q_1, q_2, q_3, q_4) \tag{14}$$

$$R = [q_5] \tag{15}$$

where $Q$ is the weighting matrices of the state error and $R$ is weighting matrices of the control signal. $q_1, q_2, q_3, q_4, q_5$ are the weight factors of the lateral error, lateral error rate, heading error, heading error rate, and the steering angle in the front wheel, respectively. Substituting Equation (11) into (13), the Lagrange multiplier approach is employed to build the constraints as follows:

$$J = \sum_{k=0}^{n-1} [X_k^T Q X_k + U_k^T R U_k + \lambda_{k+1}^T (\overline{A_k} X_k + \overline{B} U_k) - \lambda_{k+1}^T X_{k+1}] + X_n^T Q X_n \tag{16}$$

The Hamiltonian function is defined as:

$$H_k = X_k^T Q X_k + U_k^T R U_k + \lambda_{k+1}^T (\overline{A_k} X_k + \overline{B} U_k) \tag{17}$$

From Equations (16) and (17), one can obtain:

$$J = \sum_{k=0}^{n-1} [H_k - \lambda_k^T X_k] + X_n^T Q X_n + \lambda_0^T X_0 - \lambda_n^T X_n \tag{18}$$

The extreme value of Equation (18) is achieved:

$$U_k = -(R + \overline{B}^T P_{k+1} \overline{B})^{-1} \overline{B}^T P_{k+1} \overline{A_k} X_k \tag{19}$$

where K can be calculated as:

$$K = (R + \overline{B}^T P_{k+1} \overline{B})^{-1} \overline{B}^T P_{k+1} \overline{A_k} \tag{20}$$

$P$ is solved by the Riccati equation:
$$P_k = Q + \overline{A_{k+1}}^T P_{k+1} \overline{A_{k+1}} - \overline{A_{k+1}}^T P_{k+1} \overline{B} (R + \overline{B}^T P_{k+1} \overline{B})^{-1} \overline{B}^T P_{k+1} \overline{A_{k+1}}$$
Equation (19) can be rewritten as:

$$U_k = -KX_k \tag{21}$$

where K = $[K_1, K_2, K_3, K_4]$ is the gain of the LQR controller. Substituting Equation (21) into Equation (10) obtains:

$$\dot{X} = (A_k - BK)X_k + C_k \dot{\theta} \tag{22}$$

According to Equation (22), irrespective of the specific value assigned to gain K, the distance error and heading error of an automated vehicle cannot be zero during the control process, indicating the presence of a steady-state error in the system. Consequently, the influence of $C_k \dot{\theta}$ is removed and feedforward control $\delta_{ff}$ is uses as follows:

$$U = \delta_f = -KX + \delta_{ff} \tag{23}$$

Substitute Equation (23) into (10), so that when $\dot{X} = 0$, the formula for the state variable without a steady-state error is as follows:

$$X = -(A_k - BK)^{-1}(B\delta_{ff} + C_k \dot{\theta}_r) \tag{24}$$

By solving and simplifying Equation (24), the following is obtained:

$$\begin{Bmatrix} \dot{e}_d \\ \ddot{e}_d \\ \dot{e}_\varphi \\ \ddot{e}_\varphi \end{Bmatrix} = \begin{Bmatrix} \frac{1}{k_1}(\delta_{ff} - \frac{\dot{\theta}_r}{v_x}(a + b - bK_3 - \frac{mv_x^2}{a+b}(\frac{b}{C_f} + \frac{a}{C_r}K_3 - \frac{a}{C_r}))) \\ 0 \\ -\frac{\dot{\theta}_r}{v_x}(b + \frac{a}{a+b}\frac{mv_x^2}{C_r}) \\ 0 \end{Bmatrix} \tag{25}$$

Following Equation (25), when $e_d = 0$, the feedforward control is:

$$\delta_{ff} = -\frac{\dot{\theta}_r}{v_x}[a + b - bK_3 - \frac{mv_x^2}{a+b}(\frac{b}{C_f} + \frac{a}{C_r}K_3 - \frac{a}{C_r})] \tag{26}$$

In Equation (6), we assume that the real heading error can be calculated as $e_\varphi = \varphi - \theta_r$. Reducing the heading error to zero and ensuring that $e_\varphi = \varphi - \theta_r = -\beta$ are crucial when the vehicle reaches a stable state. Consequently, there is no need to devise a feedforward

controller for eliminating the steady-state error in $e_\varphi$. Additionally, Kapania et al. [25] demonstrated that achieving a steady-state equilibrium is still possible even when there are non-zero values for a lateral error as well as a heading angle error.

*2.3. Calibration Optimization*

- Calibration solution in the state of the art
  Matrix K is the gain of the LQR controller and can be calculated from Equation (19). The key to the LQR controller lies in the choice of the weight factors of matrices Q and R in Equation (14) and (15). In the previous study [41], Li et al. chose these weight factors based on the empirical method. First, Q is optimized by setting an intermediate value (arbitrarily chosen to be in the order of $10^5$) to R and Q is considered diagonal to simply tuning. Q is tested for a small value $Q_{min}$ (close to 0) and later for a large value of $Q_{max}$ (in the order of $10^6$). The two results are compared, and then an intermediate value $Q_{avg}$ is tested and the process is repeated by considering the new "small" and "large" limits $Q_{avg}$ and the value between $Q_{min}$ and $Q_{max}$ that yielded the best result. The process is then repeated by choosing a new intermediate value between the two limits until an acceptable result is produced. For R optimization, the same approach is followed, setting Q to the optimized value just found.

- Artificial Bee Colony Algorithm
  The flowchart of the ABC algorithm [42] and the LQR controller is shown in Figure 3. The employed bees actively seek K values in the vicinity of their remembered food source, all the while communicating information about these K values to the onlooker bees. The onlooker bees are likely to select good K values from those based on the fitness function as well as evaluate K values using the cost function. A few employed bees translate into the scout bees and search for new food sources until the condition is satisfied.

**Figure 3.** Flow chart of ABC and LQR controller.

- Genetic algorithm
  The GA [43,44] draws inspiration from biological principles such as mutation, crossover, and selection. The GA commences by configuring the parameters of the BMW car, the initial K values, as well as randomly generated individuals, initiating the evolutionary journey. In each generation, the fitness of every individual is evaluated, typically by assessing the value of the fitness function. Stochastic selection is employed to favor fitter individuals from the current population. These selected individuals' genomes are then subject to modifications, such as recombination and possibly random mutations, to generate a new generation of candidate solutions. This cyclic process continues, with the newly formed generation becoming the basis for the subsequent iteration. The GA advances through iterations until it reaches a termination condition. The flowchart of GA-LQR is shown in Figure 4.

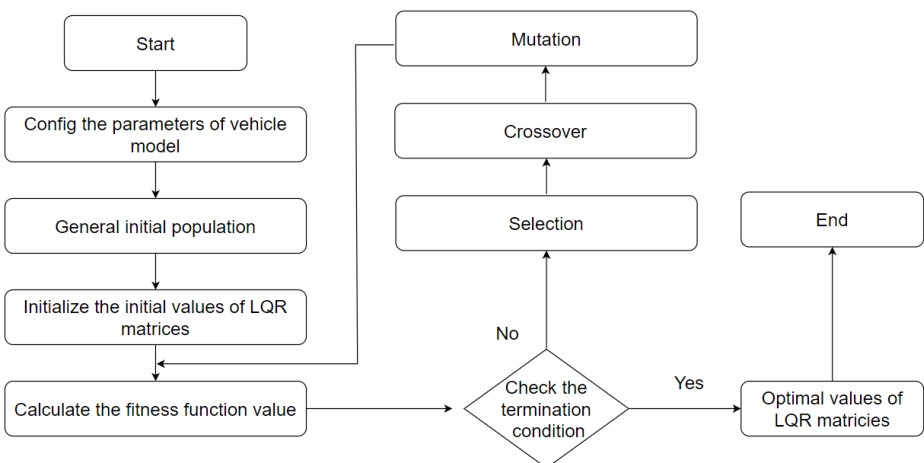

**Figure 4.** Flow chart of GA and LQR controller.

- Particle swarm optimization

  Figure 5 illustrates the flowchart of the PSO algorithm [45] and LQR controller. The optimal weight factor is found by searching for the global in order to enhance the LQR controller performance. Firstly, the parameters of the BMW car model are configured. Then, we set the initial particles' positions and velocities as well as the initial values of LQR matrices. After that, the algorithm will calculate the fitness function value and find the global best solution. If the condition is satisfied, the algorithm is stopped, otherwise, the algorithm continues to run until satisfies the condition.

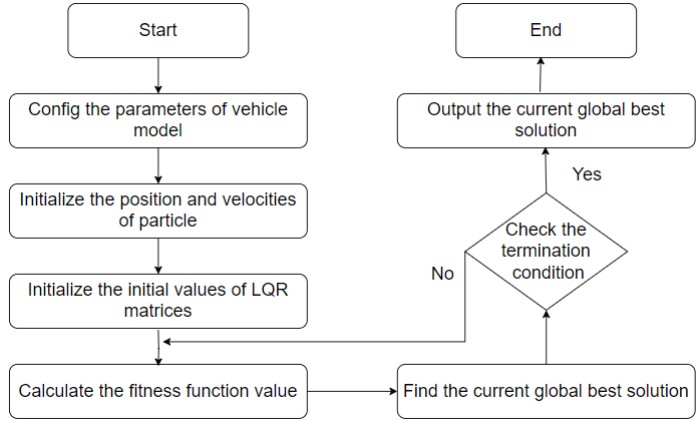

**Figure 5.** Flow chart of PSO and LQR controller.

## 3. Simulation and Results

### 3.1. Model-in-the-Loop Simulation

In this study, a BMW5 model is used for simulation. The model was calibrated with experiments performed at the Institute of Automotive Engineering laboratory and on a proving ground. Firstly, test the BMW 5 car on the real road to measure the parameters and save these in the datasheet. Then, a simple model with parameter adjustment from the datasheet is created. After that, the car was simulated on the test road, and the parameters were measured and the compared with the simulation. Finally, calibrate the steering model and stabilizer model for good fitting of the curves and choose the best tire model. The results of the model calibration refer to [46]. The BMW5 model is simulated in the CarMaker environment on a 50 m circular road, as depicted in Figure 6. The car drives in 48 s, with a maximum speed of 50 km/h and the speed step is 0.01. Therefore, there are 5000 values of matrix K corresponding to each speed step, and the lateral errors are evaluated during driving.

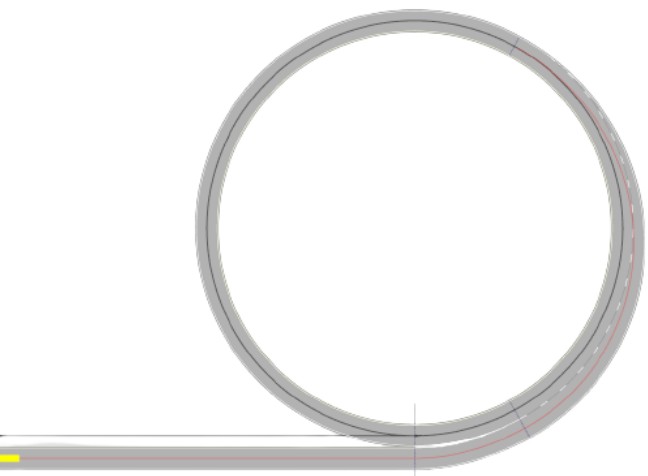

**Figure 6.** Road test for the automated vehicle.

Figure 7 and Table 1 illustrate the vehicle's steering angle and lateral error comparison under four control schemes: the manual LQR controller, the PSO based on the LQR (PSO-LQR) controller, the ABC based on LQR (ABC-LQR), and the GA based on LQR controller (GA-LQR).

**Table 1.** Lateral error comparison between optimization algorithms.

| Compare | Manual | ABC | GA | PSO |
|---|---|---|---|---|
| Absolute max values dL | 1.2026 | 2.6069 | 1.5611 | 0.0576 |
| Absolute mean values dL | 0.0249 | 0.0242 | 0.0207 | 0.0061 |
| Absolute max values L | 3.5481 | 4.4860 | 3.5151 | 0.7532 |
| Absolute mean values L | 0.5854 | 0.7696 | 0.5549 | 0.1582 |

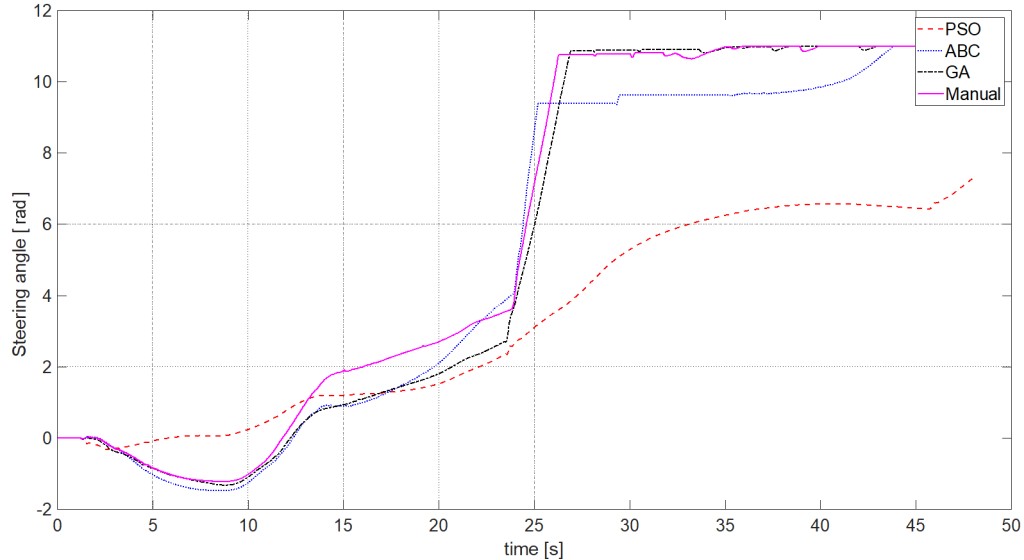

**Figure 7.** Steering angle of automated vehicle.

### 3.2. Hardware-in-the-Loop Simulation

In previous works conducted by Li et al. [30,31,47], the co-simulation framework for the virtual vehicle test bench is introduced. CarMaker and Matlab/Simulink are used together to build a co-simulation software system. The multi-body simulation (MBS) environment is provided by the CarMaker software, including vehicle dynamics, sensor models, as well as the virtual environment. The CarMaker is implemented on a real-time

processing unit called "Xpack4". Both CarMaker and Xpack 4 are from IPG Automotive GmbH [48]. Moreover, the whole vehicle test bench is controlled by the automation-system software Tornado 6 from KS Engineers [49]. The users have the capacity to configure virtual vehicle test bench using Tornado. In addition, ADAC (5 kHz) from KS Engineers [49], which is a real-time testing, controlling, and monitoring system which was developed for the virtual vehicle test bench in real-time. The EtherCAT topology protocol enables real-time synchronous simulation communication between Tornado, ADAC, and XPack4 real-time systems. In this study, the same framework is used, which comprises a virtual environment and a virtual vehicle test bench. This framework (RT-ADF) allows multiple software to be integrated and executed on real-time hardware platforms. This work demonstrates an offline virtual vehicle test bench. If the simulation model is successfully executed and validated offline, it can be transferred to the real test bench. Therefore, this is a good approach to improve the efficiency and optimize the work. Table 2 depicts the results of the vehicle model in a real-time system.

**Table 2.** Lateral error comparison between MiL and HiL.

| Compare | Absolute Max Values dL | Absolute Mean Values dL | Absolute Max Values L | Absolute Mean Values L |
|---|---|---|---|---|
| MiL | 0.0576 | 0.0061 | 0.7532 | 0.1582 |
| HiL | 0.2894 | 0.0098 | 2.5857 | 0.5339 |

In [50], a quantitative comparison method is the relative error criterion (REC) used to validate results as:

$$Error_L = \frac{|Peak_{test} - Peak_{sim}|}{|Peak_{test}|} \times 100\% = 53.23\% \tag{27}$$

$$Error_{dL} = \frac{|Peak_{test} - Peak_{sim}|}{|Peak_{test}|} \times 100\% = 78.89\% \tag{28}$$

We calculate the maximum absolute value of the error as:

$$Error_L = max(|L_{test} - L_{sim}|) = 2.35 \tag{29}$$

$$Error_{dL} = max(|dL_{test} - dL_{sim}|) = 0.28 \tag{30}$$

## 4. Discussion

Figure 7 shows the steering angle of the vehicle in four cases using manual calibration, ABC, GA, and PSO algorithms. In the interval spanning from the 4th to the 10th second, alterations in the steering angle were observed for the manual calibration, GA, and ABC algorithms, registering an approximate variation of 1.9 rad. Conversely, the steering angle adjustment remained negligible when employing the PSO algorithm. Moreover, at the 24th second, a sudden soar in the steering angle occurred within a span of 3 s, for the Manual calibration, GA, and ABC algorithms. This value then stabilized thereafter at approximately 11 rad. In contrast, the steering angle change generated by the PSO algorithm exhibited a gradual increment, culminating at 8 rad by the conclusion of the aforementioned timeframe. Therefore, the PSO algorithms can prevent the centrifugal acceleration experienced in the vehicle. Consequently, this incremental approach has the capacity to enhance both the precision of lateral displacement and the lateral velocity measurements.

Table 1 compares the lateral error and lateral velocity using three algorithms as well as manual calibration. Regarding the root mean square values for lateral displacement, the PSO algorithm exhibits a value of 0.2274 m, outperforming the manual calibration (1.124 m), ABC (1.5092 m), and GA (1.0747 m). Moreover, the results across various values

in the PSO algorithm consistently demonstrate a superior performance compared to the other algorithms under consideration.

The lateral displacement and lateral velocity differences between the MiL and HiL are evident in Table 2. Notably, simulation values obtained from the HiL are larger than those generated by the MiL. Similarly, there are errors between the results obtained from the MiL and the HiL, particularly up to 53.23%, with lateral displacement and 78.89% lateral velocity when we use the REC method to validate. Because the value of the lateral velocity is too small, this REC value is excessive. When we use the maximum absolute value method to validate the difference between the results obtained from the MiL and the HiL, the figures for lateral displacement and lateral velocity are 2.35 m and 0.28 m/s, respectively. This divergence can be caused by the cyclic delay existing between Xpack and ADAS and the computer.

## 5. Conclusions

This study presents three optimization algorithms, ABC, GA, and PSO to fine-tune the LQR parameters based on open source software for automated vehicle Apollo frameworks. These algorithms aim to overcome the reliance on empirical methods and enable a data-driven approach to LQR calibration. This study highlights the superiority of the PSO algorithm by comparing the outcomes of this optimization algorithm to the manual, ABC, GA LQR controller within the MiL simulation. Finally, the PSO algorithm is simulated on a HiL, and the result shows the gap between the MiL and the HiL simulation.

In the future, our works will continue to focus on optimizing and enhancing our system, especially focusing on the controllers and path planning, where MPC and reinforcement learning can be applied. Moreover, the more complex traffic scenarios will be used to test and validate the reality of Apollo on the test bench and real car.

**Author Contributions:** Conceptualization, D.-T.B. and H.L.; methodology, D.-T.B. and H.L.; software, D.-T.B. and H.L.; validation, D.-T.B., H.L. and F.D.C.; formal analysis, D.-T.B. and H.L.; investigation, D.-T.B. and H.L.; resources, D.-T.B. and H.L.; data curation, D.-T.B. and H.L.; writing—original draft preparation, D.-T.B. and H.L.; writing—review and editing, D.-T.B., H.L., F.D.C. and A.E.; visualization, D.-T.B. and H.L.; supervision, A.E.; project administration, A.E.; funding acquisition, A.E. All authors have read and agreed to the published version of the manuscript.

**Funding:** Funding by the Graz University of Technology. This activity is part of the research project InVADE (FFG nr. 889349) and has received funding from the program Mobility of the Future, operated by the Austrian research funding agency FFG. Mobility of the Future is a mission-oriented research and development program to help Austria create a transport system designed to meet future mobility and social challenges. This paper is supported by TU Graz Open Access Publishing Fund.

**Institutional Review Board Statement:** Not applicable.

**Informed Consent Statement:** Not applicable.

**Data Availability Statement:** The data presented in this study are available in Master's Thesis "Aufbau von Validierten Fahrzeugmodellen", Graz University of Technology, Austria, 2023.

**Acknowledgments:** We extend our gratitude to Zhengguo Gu for their valuable internship support, which greatly contributed to the success of this project.

**Conflicts of Interest:** The authors declare that the research was conducted in the absence of any commercial or financial relationships that could be construed as a potential conflict of interest. Additionally, the authors declare that this study received funding from the Austrian research funding agency FFG within the scope of the research project InVADE. The funder was not involved in the study design, collection, analysis, interpretation of data, the writing of this article, or the decision to submit it for publication.

## Nomenclature

| | |
|---|---|
| $\alpha_f$ | steering angle in the front wheel |
| $\alpha_r$ | steering angle in the rear wheel |
| $\delta_f$ | front steering angle |
| $\delta_r$ | rear steering angle |
| $O$ | mass center point |
| $a$ | distance from O to the front axle |
| $b$ | distance from O to the rear axle |
| $\varphi$ | yaw angle |
| $\dot{\varphi}$ | yaw rate angle |
| $C_f$ | lateral stiffness of the tires in the front wheels |
| $C_r$ | lateral stiffness of the tires in the rear wheels |
| $F_{y_f}$ | the lateral force of the front wheels |
| $F_{y_r}$ | the lateral force of the rear wheels |
| $v_x$ | longitudinal velocity |
| $v_y$ | lateral velocity |
| $m$ | mass of the vehicle |
| $I_z$ | rotation inertia of vehicle around the vertical axis |
| $e_d$ | lateral error |
| $e_\theta$ | heading error |
| $\theta$ | heading angle of the vehicle |
| $\theta_r$ | reference heading angle |
| $\beta$ | side slip angle |
| $\dot{e}_d$ | the first-order derivatives of the lateral error |
| $\dot{e}_\varphi$ | the first-order derivatives of the heading angle error |

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
