# Peer review of "Lateral Control Calibration and Testing in a Co-Simulation Framework for Automated Vehicles"

_applsci, doi:10.3390/app132312898_

Round 1

Reviewer 1 Report

Comments and Suggestions for Authors

The authors of this paper proposed an LQR Controller to control automated vehicles (Apollo frameworks). They used three optimization techniques to to tune the parameters, mainly, ABC, PSO and GA.

For the purpose of digital simulation, the disseized the continuous model and derived the results for the discrete one.

The work is very good, but it can be improved according to the following comments.

1.    The authors did not state clearly what are the parameters to be found. Are they the weight Q and R plus the state feedback gain K, or only Q and R?

2.    The authors did not list in a table the final values of Q, R and K they obtained from optimization and used in the simulation.

3.    Figure 1 was taken from the Literature. It should be cited.

4.    In Equation (10), the notation for the control signal is different than that was written in the next line, 147. This is repeated in many Equations next.

5.    Use subscript in the symbols in line 147 and other lines.

6.    In Equation (12), dt is used for sampling time. Which is not usual. dt is used for the derivative.

7.    Equation (19) contains P_k+1, but the subscript (k+1) in the Ricatti equation next is dropped.

8.    No references were added inside the text describing ABC and GA algorism in page 7.

Comments on the Quality of English Language

Minor editing,

Reviewer 2 Report

Comments and Suggestions for Authors

This manuscript introduces three optimization algorithms—ABC, GA, and PSO—that are used to refine LQR parameters in autonomous vehicle Apollo systems built on open source software. The objective of these algorithms is to eliminate the need for empirical approaches and provide a data-driven approach to the LQR calibration. In my opinion the manuscript is well-organized and contain the needed contributions to be published. However some points in the results return the article to be revised before the final acceptance decision. First, one example of a circle road is tackled where in the theoretical basis of the manuscript the authors can tackle more than one case like spiral road case. Second, Fig. 7 report only the decision angle of the steering wheel of the studied automated vehicle where you can also control other dynamics like accelerating and decelerating. Third, application of the GA in the field of transportation planning is absent like: Complete hierarchical multi-objective genetic algorithm for transit network design problem and Multi-objective transit route network design as set covering problem.

Reviewer 3 Report

Comments and Suggestions for Authors

The scientific article titled: “Lateral Control Calibration and Testing in a Co-simulation Framework for Automated Vehicles” raises an important issue regarding calibration and control in vehicles. The use of different optimization algorithms, Colony Optimization, Genetic Algorithm and Particle Swarm Optimization, is interesting. The advantage of the article is the presentation of the patterns used, but the disadvantage is the lack of a list of the symbols used. The results obtained were presented clearly. Figure 2 should be larger because it is difficult to read at this scale. The literature includes new research works, but several older items are missing. The summary of the work shows further plans for the development of the system, which is considered an advantage.

Round 2

Reviewer 2 Report

Comments and Suggestions for Authors

None.

Comments on the Quality of English Language

None.
